# Effect of *Ferulago lutea* (Poir.) Grande Essential Oil on Molecular Hallmarks of Skin Aging

**DOI:** 10.3390/plants12213741

**Published:** 2023-10-31

**Authors:** Jorge M. Alves-Silva, Patrícia Moreira, Carlos Cavaleiro, Cláudia Pereira, Maria Teresa Cruz, Lígia Salgueiro

**Affiliations:** 1Univ Coimbra, Institute for Clinical and Biomedical Research, Health Sciences Campus, Azinhaga de S. Comba, 3000-548 Coimbra, Portugal; jmasilva@student.ff.uc.pt; 2Univ Coimbra, Faculty of Pharmacy, Health Sciences Campus, Azinhaga de S. Comba, 3000-548 Coimbra, Portugal; cavaleir@ff.uc.pt; 3Univ Coimbra, Center for Innovative Biomedicine and Biotechnology, 3000-548 Coimbra, Portugal; patriciaraquel_jm@hotmail.com (P.M.); cpereira@fmed.uc.pt (C.P.); 4Univ Coimbra, Center for Neuroscience and Cell Biology, Faculty of Medicine, Rua Larga, 3004-504 Coimbra, Portugal; trosete@ff.uc.pt; 5Univ Coimbra, Chemical Process Engineering and Forest Products Research Centre, Department of Chemical Engineering, Faculty of Sciences and Technology, 3030-790 Coimbra, Portugal; 6Univ Coimbra, Faculty of Medicine, Health Sciences Campus, Azinhaga de S. Comba, 3000-548 Coimbra, Portugal

**Keywords:** lipogenesis, depigmenting activity, aging, senescence, wound healing

## Abstract

With the increase in global life expectancy, maintaining health into old age becomes a challenge, and research has thus concentrated on various strategies which aimed to mitigate the effects of skin aging. Aromatic plants stand out as promising sources of anti-aging compounds due to their secondary metabolites, particularly essential oils (EOs). The aim of this study was to ascribe to *Ferulago lutea* EO several biological activities that could be useful in the context of skin aging. The EO was obtained using hydrodistillation and characterized by gas chromatography–mass spectrometry (GC/MS). The anti-inflammatory potential was assessed using lipopolysaccharide (LPS)-stimulated macrophages. The effect on cell migration was disclosed using scratch wound assay. Lipogenesis was induced using T0901317, hyperpigmentation with 3-isobutyl-1-methylxantine (IBMX) and senescence with etoposide. Our results show that the EO was characterized mainly by α-pinene and limonene. The EO was able to decrease nitric oxide (NO) release as well as iNOS and pro-IL-1β protein levels. The EO promoted wound healing while decreasing lipogenesis and having depigmenting effects. The EO also reduced senescence-associated β-galactosidase, p21/p53 protein levels and the nuclear accumulation of γH2AX. Overall, our study highlights the properties of *F. lutea* EO that make it a compelling candidate for dermocosmetics applications.

## 1. Introduction

In 2015, the World Health Organization World Report on Aging and Health emphasized the pressing need for strategies promoting healthy aging. It specifically underscored the importance of focusing more significantly on skin health as a means of maintaining overall well-being across the entire lifespan [1]. Skin aging is induced and sustained by chronological aging and photoaging, and is clinically characterized by pigmentation, atrophy, a loss of elasticity and an impaired recovery response against damage, leading to pathologic skin disorders [2]. For instance, cutaneous infections are common among the elderly due to a reduction in the skin’s barrier function and its immune defenses. Additionally, impaired wound healing further escalates the risk of infection. It is also widely recognized that senescent cells accumulate in the skin during aging, and even though they cannot proliferate, they continue to be metabolically active. These cells display an altered secretory profile, known as a senescence-associated secretory phenotype (SASP), which includes proinflammatory cytokines that significantly modify the skin’s microenvironment and contribute to inflammaging [2]. More recently, aging-associated skin pigmentation has been actively considered, with cellular senescence playing a key role in this process [3]. With the rise in life expectancy and a growing demand for solutions to reduce the signs of aging skin, there is a heightened interest in this research field, particularly with natural cosmetic products taking the spotlight [4].

The significance of natural products is further fueled by consumers’ increasing preference for more environmentally friendly and ecologically conscious products [5]. Furthermore, the research in cosmetics aims not only to improve skin appearance during aging but also to improve the quality of life of individuals by preventing or treating skin-related diseases [4]. In this context, the term “cosmeceutical” has been coined, and is defined as a cosmetic product that includes active ingredients with drug-like benefits [5]. Natural products, including plants and their metabolites, have been used as skin care products for millennia, and modern formulations are causing their relevance to reemerge. Plant extracts are of particular interest since they are able to accomplish two tasks at once, skin care and providing a source of nutrients required for the maintenance of skin functions, thus maintaining a healthy skin [5]. This dual function is probably due to their phytochemicals, such as terpenoids, polyphenols, essential oils, and vitamins, that altogether exert a plethora of functions such as antioxidant, tyrosinase inhibition, antimicrobial, anti-inflammatory and UV radiation protection, thus mitigating several hallmarks of skin aging [4,5].

The genus *Ferulago* W.D.J. Koch comprises 48 subordinated taxa [6] that are predominantly found in mild climate zones, such as Europe, southwest and middle Asia, the Caucasus and north Africa. Although in the Iberian Peninsula, this genus is represented by four species (*F. galbanifera* (Mill.) W.D.J. Koch; *F. brachyloba* Boiss. and Reut.; *F. lutea* (Poir.) Grande and *F. granatensis* Boiss.) [7], in Portugal only the endemic species *F. lutea* (*Ferulago capillaris* (Link ex Spreng.) Cout.) is found [8]. Plants from this genus are widely used in traditional medicines as antiseptic, for wound healing, for bronchitis and as an immunostimulant [7]. As reviewed elsewhere, the biological properties of plants belonging to the *Ferulago* genus have been thoroughly investigated. Nonetheless, when comparing *F. lutea* to other species, studies on this particular plant are notably scarce. [7]. For instance, the antifungal and antibacterial potential of the essential oil from the flowers [9] and roots [10] from *Ferulago lutea* growing in Tunisia was previously reported. The Algerian *F. lutea* also presents strong antimicrobial activity [11]. The essential oil from the aerial parts of *F. capillaris* (=syn *F. lutea*) growing in Portugal also displays strong antifungal properties against several pathogenic yeasts and filamentous fungi [8]. In addition, several studies have shown that *F. lutea* exerts anti-acetylcholinesterase activity [9,10] and antioxidant properties [11]. The documented effects of *F. lutea* underscore its potential for further utilization in the context of dermocosmetics and the development of skincare formulations.

Having this in mind, the present work aims to explore the protective role of *F. lutea* essential oil collected in Portugal against skin aging hallmarks, particularly inflammation, impaired wound healing, pigmentation, and senescence. To the best knowledge of the authors, the selected bioactivities have never been explored for this species. Our findings reveal that the essential oil extracted from *F. lutea* plants in Portugal exhibits several noteworthy properties. It demonstrates anti-inflammatory effects through the modulation of the NF-κB pathway, supports wound healing, has anti-lipogenic properties and exhibits depigmenting capabilities. Additionally, our research indicates that the essential oil possesses anti-senescence properties, as it prevents the nuclear accumulation of γ-H2AX and regulates the p53/p21 signaling pathway.

## 2. Results

### 2.1. Chemical Composition

The hydrodistillation of *F. lutea* growing in Portugal produced essential oil with a yield of 0.8% (*v*/*w*). Monoterpenes hydrocarbons were the principal class of compounds detected by GC and GC/MS analysis (89.2%), with α-pinene (36.5%) and limonene (31.2%) being the major compounds (Table 1).

### 2.2. Effect of F. lutea on Cell Viability

With the potential clinical application of *F. lutea* essential oil in mind, we evaluated its impact on various cell lines that are representative of the skin, namely melanocytes, macrophages, keratinocytes, and fibroblasts. As shown in Figure 1, the dose of 1.25 µL/mL is toxic towards all tested cell lines, particularly macrophage (RAW 264.7) (Figure 1A) and fibroblast (NIH/3T3) (Figure 1B) cells. For NIH/3T3 cells, the concentration of 0.64 µL/mL also presented significant toxicity. Having these results in consideration, the dose of 0.64 µL/mL was selected for the anti-inflammatory assays; in contrast, for experiments involving NIH/3T3, HaCaT and B16V cell lines, a dosage of 0.32 µL/mL was employed.

### 2.3. The Essential Oil of F. lutea Exerts Anti-Inflammatory Effect via Inhibition of NF-κB Pathway

Considering that several plants from the genus *Ferulago* are widely used as anti-inflammatory [7], we first started by addressing if the essential oil from *F. lutea* could also present anti-inflammatory potential. To achieve this, we utilized LPS-stimulated macrophages, which trigger the activation of the NF-κB signaling pathway, resulting in the synthesis of several pro-inflammatory mediators, including IL-1β and inducible nitric oxide synthase (iNOS). iNOS, in turn, promotes the production of nitric oxide (NO). The results achieved demonstrated that the essential oil decreased NO production in a dose-dependent manner (IC_50_ = 0.653 µL/mL) (Figure 2A).

We also aimed to explore whether the reported decrease in NO could be attributed to the modulation of the transcription factor NF-κB. In order to disclose the effect of the EO on this signaling pathway, we assessed the protein levels of two pro-inflammatory mediators associated with this pathway, namely iNOS and pro-IL-1β [12]. As shown in Figure 2B,C, the presence of the EO at 0.64 µL/mL significantly decreased in the levels of both proteins, thus suggesting that the EO is able to inhibit the NF-κB cascade.

### 2.4. Ferulago lutea Essential Oil Promotes Cell Migration

Given that that aged individuals frequently experience impaired wound healing, resulting in chronic wounds and posing significant medical challenges, often in the context of a pro-inflammatory environment [13], we further assess whether the essential oil derived from *F. lutea* could stimulate fibroblast migration. As depicted in Figure 3, the essential oil increased cell migration at a concentration of 0.32 µL/mL. However, at the other tested concentrations, the promotional effect on cell migration was not observed.

### 2.5. Ferulago lutea Essential Oil Affects Lipogenesis Differentially

Given that, on one hand, lipogenesis decreases with age [14], and on the other hand, this process is associated with a higher incidence and more severe forms of acne vulgaris [15], our objective was to determine whether *F. lutea* essential oil could regulate lipogenesis in the HaCaT keratinocyte cell line. As observed in Figure 4, the presence of the essential oil affects lipogenesis differentially. Indeed, in basal conditions, the essential oil slightly stimulates lipogenesis, but not significantly. However, in the presence of T0901317, a liver X receptor (LXR) ligand known to induce lipogenesis [16], pre-treatment with the essential oil led to a significant decrease in Oil Red content as observed by microscopy analysis (Figure 4A,B) and prior to quantification (Figure 4C), indicative of lipogenesis decrease.

### 2.6. Ferulago lutea Essential Oil Has Depigmenting Properties

Even though active melanocytes decrease with skin aging, it is worth noting that hyperpigmentation is the predominant characteristic of older sun-exposed skin [17]. Indeed, abnormal pigmentation is a common symptom accompanying aging skin, such as mottled pigmentation (senile lentigo) and melasma. Considering the beneficial effects of *F. lutea* on wound healing and lipogenesis, we then wondered whether the essential oil could modulate hyperpigmentation in B16V melanocytes. As expected, the use of 3-isobutyl-1-methylxanthine (IBMX), an inducer of skin hyperpigmentation, led to an increase in both the melanin content and the activity of tyrosinase, a crucial enzyme involved in melanogenesis (Figure 5). The presence of the essential oil (0.32 µL/mL) led to a decrease in both parameters, thus suggesting that the essential oil possesses depigmenting properties.

### 2.7. Ferulago lutea Essential Oil Has Anti-Senescence Properties

Having in mind the potential of *F. lutea* essential oil on several age-related skin dysfunctions, we then assessed whether the essential oil could have a direct effect on cell senescence. For that, we used three complementary techniques, namely measuring the activity of senescence-associated β-galactosidase activity, the nuclear accumulation of yH2AX histone and the protein levels of p21 and p53, all key players in cellular senescence [17].

Regarding the activity of β-galactosidase, we observed that etoposide led to the activation of this enzyme, as observed in the increased amount of X-galactose (X-gal; Figure 6). However, when the essential oil was added in the recovery phase, the number of X-gal positive cells was significantly decreased, thus suggesting that *F. lutea* essential oil could exert anti-senescent properties.

Considering that *F. lutea* essential oil was able to decrease senescence-associated β-galactosidase activity, we then proceeded to elucidate the underlying mechanism of action by assessing the protein levels of p21 and p53, key players in senescence via the p53/p21 axis, and the nuclear accumulation of phosphorylated H2AX (γH2AX). As observed in Figure 7, the presence of the essential oil greatly decreases the phosphorylation of H2AX, thus validating the anti-senescence potential of the essential oil. Considering that nuclear accumulation of γH2AX leads to cell cycle arrest via p53/p21 signaling pathway, we proceeded to assess the effect of the essential oil on the expression of these markers. As expected, the etoposide-only treated cells had an accumulation of both p53 and p21. A significant decrease in both p53 and p21 protein levels was observed in the presence of the essential oil during the recovery phase (Figure 8), thus suggesting that *F. lutea* essential oil prevents cell cycle arrest in the presence of DNA damage.

## 3. Discussion

The current study underscores the positive effect of *F. lutea* essential oil on skin aging hallmarks. We have demonstrated that this essential oil possesses anti-inflammatory properties, supports wound healing, reduces lipogenesis, exhibits depigmenting abilities, and displays anti-senescent properties. Furthermore, our results demonstrated that the chemical composition of *F. lutea* from Portugal is consistent with earlier studies on this species [8]. This underscores that the essential oil extracted from plants growing in Portugal is primarily characterized by the presence of α-pinene and limonene. Overall, we suggest that *F. lutea* essential oil constitutes a likely promising source for prioritizing molecules with anti-aging properties.

A pro-inflammatory cytokine production in the skin is often associated with the onset and progression of several skin disorders [18]. Undoubtedly, the cutaneous immune response plays a critical role in the regulation of skin aging and the development of immune-mediated skin conditions, notably eczema, acne, atopic dermatitis, and psoriasis [19,20]. Because of the adverse effects reported for the conventional therapies (glucocorticoids and immunosuppressants), research has focused on the development of new therapeutic alternatives [21,22]. Based on the reported findings, we propose that *F. lutea* EO may hold promise in the management of inflammatory skin conditions. While specific studies regarding the anti-inflammatory potential of *F. lutea* are still pending, it is worth noting that one study showed that *F. campestris*, which is rich in myrcene, α-pinene and γ-terpinene, reduced mRNA levels of IL-1β, IL-6 and iNOS. [23]. The anti-inflammatory effect might be attributed to the high amounts of α-pinene and limonene. Indeed, a study has shown that both major compounds from *F. lutea* exhibit anti-inflammatory effects in an acute model of pancreatitis [24]. Other studies have also demonstrated the anti-inflammatory potential of limonene [25,26] and α-pinene [27,28,29], thereby confirming their roles in the activity of *F. lutea* essential oil.

Skin aging often leads to impaired wound healing due to the prolongation of the inflammatory phase in the wound-healing process [30], thus contributing to the emergence of chronic wounds [13]. Considering that the essential oil from *F. lutea* exerts anti-inflammatory properties, we then hypothesize that it could also contribute beneficially to the wound-healing process. Indeed, our results show that the essential oil promotes wound healing in a dose-dependent manner. Regarding studies with plants from the genus *Ferulago*, no studies have been conducted thus far; however, some studies have shown that the major compounds from the essential oil reported herein have wound-healing potential. For instance, α-pinene has been shown to promote the formation of stress-resistant scars and wound contractions by eliciting collagen deposition [31]. Other studies have also demonstrated wound-healing properties for limonene [32,33,34], including in a diabetic animal model, where a decrease in the pro-inflammatory environment was also observed [33], similarly to our present study, thus suggesting that limonene might be the biggest contributor to the wound-healing promotion capability of *F. lutea* growing in Portugal.

During skin aging, sebaceous cells in the skin lose their functionality, leading to a decrease in surface lipid levels and xerosis [35]. In contrast, acne vulgaris promotes the production of sebum by activating the enzyme acetyl-CoA carboxylase (ACC) [15]. Our findings indicate that *F. lutea* essential oil may offer advantageous outcomes in mitigating the decline in lipogenic capabilities associated with aging skin. Additionally, when applied in conditions characterized by elevated lipogenesis, such as acne vulgaris, it could potentially alleviate excessive sebum production. To the best of the authors’ knowledge, there is a lack of studies conducted on the impact of other *Ferulago* species on lipogenesis.

Regarding isolated compounds, only limonene has been explored in lipogenesis. A study demonstrated that limonene exhibited a dual effect on 3T3-L1 adipocytes. On the one hand, it increased lipid accumulation in adipocytes by activating the PPARγ, C/EBP-α and C/EBP-β pathways. On the other hand, at higher doses, it had an opposite effect by reducing lipid accumulation [36]. In another study, limonene was able to modulate AMPK-mediated expression of mRNA genes related to adipogenesis (PPARγ, C/EBPα and FABP4) and lipogenesis (SREBP-1c, ACC and FAS) [37]. Having these results in mind, we suggest that limonene might be the major contributor to the anti-lipogenic effects reported herein for *F. lutea* essential oil.

Although the number of melanocytes decrease, abnormal pigmentation is a common symptom accompanying aging skin. Furthermore, chronic UV exposure also leads to the loss of function in fibroblasts, which help in the regulation of melanogenesis [38]. Indeed, the communication between melanocytes as pigmentary cells and other adjacent types of aged skin cells such as senescent fibroblasts contribute to the aged-skin-associated pigmentation. Our results show that the presence of *F. lutea* essential oils decreases melanin, probably due to the inhibitory effect on tyrosinase, a key player in melanin production [39]. Our study is the first addressing the depigmenting properties of extracts obtained from plants from the genus *Ferulago*. Regarding isolated compounds, few studies have been conducted using limonene or α-pinene. Limonene was shown to protect keratinocytes against UVB-induced photodamage and photoaging [40]. In addition, this compound is part of the commercial product Garnier Dark Spot Corrector [41], thus reinforcing its skin lightening properties.

It is known that senescent cells contribute to skin aging [42,43], thus leading to the emergence of aged-skin-related pathologies. Indeed, it has been shown that fibroblasts exposed to UVB exhibit DNA damage and consequent cell cycle arrest, in addition to the expression of senescence markers, particularly senescence-associated β-galactosidase activity and p16, p21 and p53 activation [42]. Importantly, senescent cells display an altered SASP, which includes proinflammatory cytokines. Our results show that *F. lutea* essential oil reduces the activity of senescence-associated β-galactosidase, thus suggesting anti-senescence properties for this oil. The accumulation of γH2AX is a response to damaged DNA that ultimately leads to cell cycle arrest and cellular senescence [44]. Furthermore, as reviewed elsewhere [45], double strand breaks only lead to cellular senescence when they are resistant to repair processes, leading to a subset of γH2AX called persistent γH2AX. Having in mind that the essential oil decreases the accumulation of this marker, we hypothesize that it could prevent more serious damage to the DNA, thus allowing its repair, which in turn prevent the activation of the p53/p21 axis for cell cycle arrest and consequent cellular senescence. Indeed, our results demonstrate that *F. lutea* essential oil reduces the protein levels of both p53 and p21, thus reinforcing the anti-senescent potential of the essential oil. Our study highlights for the first-time anti-senescence properties for plants from the genus *Ferulago*. The reported activity might be attributed to the presence of limonene and α-pinene. Indeed, limonene protects HaCaT cells from H_2_O_2_-induced aging [46] and protects human keratinocytes from UVB-induced photoaging by activating Nrf2-dependent antioxidants [40]. UVA-induced photoaging was also prevented by α-pinene through the inhibition of matrix metalloproteinase expression [47].

Overall, our study is the first to point out several properties of *F. lutea* essential oil that are relevant in the management of skin aging and associated pathologies. However, further studies are needed in order to further validate these effects using more physiologically relevant models.

## 4. Materials and Methods

### 4.1. Plant Material and Essential Oil Distillation

Umbels with mature seeds of *F. lutea* were collected on 22 August 2022 in Central Portugal (Celorico da Beira). Species authenticity was determined by Jorge Paiva, a taxonomist of the University of Coimbra. A voucher specimen was deposited at the Herbarium of University of Coimbra. Prior to the hydrodistillation, the plant material was air-dried in the dark.

The essential oil was obtained through hydrodistillation using a Clevenger-type apparatus, for 3 h, as recommended by the European Pharmacopoeia [48].

### 4.2. Chemical Characterization of the Essential Oil

The essential oil was analyzed by gas chromatography (GC) and gas chromatography coupled to mass spectrometry (GC/MS), as previously described by our group [49].

The identification of the volatile compounds was achieved by analyzing their retention indices (RIs) on two GC columns (SPB-1 and SupelcoWax-10 (Sigma-Aldrich, St. Louis, MO, USA)) and mass spectra. RIs were matched with those from a database made by us and/or from literature data [50,51,52,53]. Mass spectra were compared with reference spectra from our own database, Wiley/NIST library [54] and literature data [50,55].

### 4.3. Cell Culture

The cell lines RAW 264.7 (mouse leukemic macrophage cell line, ATCC TIB-71) and NIH 3T3 (mouse embryonic fibroblast, ATCC CRL-1658) were obtained from American Type Culture Collection and cultured as previously described [56]. The cell lines HaCaT (human keratinocyte cell line) and B-16V (mouse melanoma cell line) were obtained from Cell Line Services (CLS 3004993, Eppelheim, Germany) and German Collection of Microorganisms and Cell Cultures (DSMZ ACC-370, Braunschweig, Germany), respectively, and were culture as previously reported [57].

### 4.4. Effect on Cell Viability

The effect of different concentrations of the essential oil on the viability of macrophages, fibroblasts, keratinocytes, and melanocytes was evaluated through the resazurin reduction test [58]. RAW 264.7 macrophages (0.6 × 10^6^ cells/mL), NIH/3T3 fibroblasts (5 × 10^4^ cells/mL), HaCaT keratinocytes (1 × 10^5^ cells/mL) and B-16V melanocytes (6 × 10^4^ cells/mL) were seeded in 48-well plates. After a 24 h incubation with the essential oil (0.08–1.25 µL/mL), cell viability was determined as previously reported [58].

### 4.5. Anti-Inflammatory Potential

#### 4.5.1. Nitric Oxide Production

The capacity of the essential oil to decrease the nitric oxide production in lipopolysaccharide (LPS)-stimulated macrophages was assessed using the methodology described in our group [49,59].

#### 4.5.2. Western Blot Analysis of Pro-Inflammatory Mediators

RAW 264.7 cells (0.8 × 10^6^ cells/well) were cultured in 6-well plates and stabilized overnight. Cells were then subjected to 1 h incubation with the essential oil (0.64 µL/mL), followed by 24 h of LPS activation (50 ng/mL). Negative and positive controls comprising untreated and LPS-only-treated cells, respectively, were included. Cell lysate preparation followed the protocol previously described in Zuzarte et al. [60].

The protein levels of the inducible nitric oxide synthase (iNOS) and IL-1β pro form (pro-IL-1β), with tubulin used as loading control, were assessed using Western blot, as previously described [61].

### 4.6. Cell Migration

The effect of the extract on cell migration was investigated through the scratch wound assay according to Martinotti et al. [62] with slight modifications. Briefly, after the scratch induction, NIH/3T3 fibroblasts were incubated in DMEM with 2% FBS alone with the addition of the essential oil (0.08–0.32 µL/mL) for 18 h. After image acquisition, the open area was quantified as reported by our team [61].

### 4.7. Inhibition of Lipogenesis

#### 4.7.1. Lipogenesis Induction

Lipogenesis was induced using T0901317, an activator of the LXR [63]. Briefly, HaCaT cells were seeded in a 12-well plate at 3 × 10^4^ cells/mL and left to adhere for 24 h. Afterwards, cells were pre-treated with 0.32 µL/mL of the essential oil for 1 h before being stimulated with T0901317 for 24 h. At the end, medium was removed, and cells were washed three times with PBS and then fixated for 30 min with 4% paraformaldehyde at room temperature. Then, cells were washed with PBS and stored at 4 °C until further analysis.

#### 4.7.2. Oil Red-O Fluorescent Staining

After cell fixation, cells were stained with Oil Red O dye (6:4, 0.6% Oil Red-O dye in water) for 15 min. Cells were washed three times with PBS and then images were acquired using a bright-field microscope and Oil Red O staining was quantified using the Threshold function in image J/Fiji ver. 1.54f software.

#### 4.7.3. Lipid Accumulation Quantification by Oil Red O Staining

After the procedure described in Section 4.7.2, Oil Red O staining was dissolved in 200 µL of isopropanol, the contents of each well were transferred to a new 96-well plate, and the absorbance was measured at 500 nm using an automated plate reader. Oil Red O content in each condition was determined using the following equation, normalized to control conditions:Oil Red O content = Abs_t_/Abs_CT_
where Abs_t_ is the absorbance at 500 nm of each condition, and Abs_CT_ is the absorbance at 500 nm in control cells (untreated cells).

### 4.8. Depigmenting Effect

The depigmenting activity was determined through the methodology previously described [57]. Briefly, B-16V cells (6 × 10^5^ cells/mL, triplicates) were seeded in 6-well plates and allowed to adhere for 24 h. Afterwards, the cells were incubated for 48 h, in culture medium alone, or were treated with 3-isobutyl-1-methylxanthine (IBMX, 200 µM) (Sigma-Aldrich, St. Louis, MO, USA), a well-known inducer of skin pigmentation, in the presence or absence of 0.32 µL/mL of the EO. At the end of the incubation period, the cells were washed twice with ice-cold PBS, lysed, and scrapped in ice-cold lysis buffer (50 mM sodium phosphate (pH 6.5), 1% (*v*/*v*) Triton X-100, 0.1 mM phenylmethylsulphonyl fluoride (PMSF), 1 mM EDTA). The cell lysates were kept at −80 °C for 30 min, defrosted and then centrifuged at 12,000× *g* for 10 min at 4 °C; finally, the supernatants were collected for tyrosinase activity analysis. The protein content in the supernatants was determined using the bicinchoninic acid protein assay. The pellets were dissolved in 200 µL 1 N NaOH for 1 h at 95 °C, then 100 µL were transferred to a 96-well plate, and the absorbance was measured at 400 nm using an automated plate reader to determine their melanin content, which was normalized to the protein content.

For tyrosinase activity, the supernatants were transferred to a 96-well plate with 2.5 mM L-3,4-Dihydroxyphenylalanine (L-DOPA), a commercially available tyrosinase substrate (#D1507, Sigma-Aldrich, St. Louis, MO, USA). The tyrosinase activity was measured at 37 °C at 475 nm, at 5 min time intervals for 1 h, with automated plate reader. The tyrosinase activity was normalized to their protein content.

### 4.9. Anti-Senescence Potential

#### 4.9.1. Senescence-Associated β-Galactosidase Activity

Senescence was evaluated using the senescence inducer etoposide, as reported elsewhere [57].

#### 4.9.2. yH2AX Staining

For the nuclear staining of histone yH2AX, NIH/3T3 fibroblasts were seeded at 1 × 10^5^ cells/mL in glass coverslips and treated as reported in Section 4.9.1. At the end of the treatment, cells were fixed with 4% paraformaldehyde for 15 min, followed by three washes with sterile PBS. Then, cells were permeabilized with 0.1% Triton X-100 for 15 min and washed with PBS (three times). Cells were incubated with blocking solution (3% bovine serum albumin, 10% goat serum in PBS) for 1 h. Then, the primary antibody against yH2AX (1:500, #9718 Cell Signalling, Danvers, MA, USA), which was prepared in blocking solution, was added and the cells were incubated overnight at 4 °C. Afterwards, coverslips were washed with PBS (three times), and incubated for 1 h at room temperature with the corresponding secondary antibody (1:500, goat anti-rabbit Alexa Fluor 564) and DAPI (1:2000) prepared in blocking solution. After washes with PBS, coverslips were mounted in glass slides with Mowiol mounting medium. Images were acquired in a confocal point-scanning microscope (Zeiss LSM710; Carl Zeiss, Oberkochen, Germany) in 40× objective.

#### 4.9.3. p21 and p53 Protein Levels

NIH/3T3 fibroblasts (2.5 × 10^5^ cells/mL) were plated in 6-well plates and then treated as reported in Section 4.9.1. At the end, cell lysates were prepared as reported by Zuzarte et al. [60]. Protein separation and immunoblotting were performed as previously reported in our group [64]. After the blocking step, membranes were incubated with specific p21 antibody (1:1000, Abcam ab188224) and p53 (1:1000, Proteintech 10442-1-AP) overnight at 4 °C. Then, membranes were washed with TBS-T followed by 1 h incubation at room temperature with horseradish peroxidase-conjugated secondary antibodies (1:20,000). Then proteins were detected with a chemiluminescence scanner (Image Quant LAS 500, GE Life Sciences, Marlborough, MA, USA).

### 4.10. Statistical Analysis

The experiments were performed at least in duplicate for at least three independent experiments. Mean values ± SEM (standard error of the mean) are presented in the results. Statistical significance was evaluated using one-way analysis of variance (ANOVA) or Mann–Whitney test followed by the appropriate post hoc test analysis using GraphPad Prism version 9.3.0. *p* values < 0.05 were accepted as statistically significant.

## 5. Conclusions

The present study highlights several activities of the essential oil of *F. lutea* growing in Portugal that are particularly relevant in the context of skin aging and skin-related pathologies. We demonstrate that the essential oil exerts its anti-inflammatory effects probably via the modulation of the NF-κB signaling pathway. Furthermore, we report wound-healing promotion, anti-lipogenic and depigmenting properties of the essential oil, thus highlighting its relevance in skin aging and associated pathologies. We hypothesize that these effects might be due to the anti-senescence properties that we report for the essential oil, due to the modulation of the p21/p53 pathway.

Overall, these results promote the industrial interest in *Ferulago lutea*, particularly for dermocosmetics. In addition, they brings new insights into the management of skin aging and associated pathologies using natural products.

## Figures and Tables

**Figure 1 plants-12-03741-f001:**
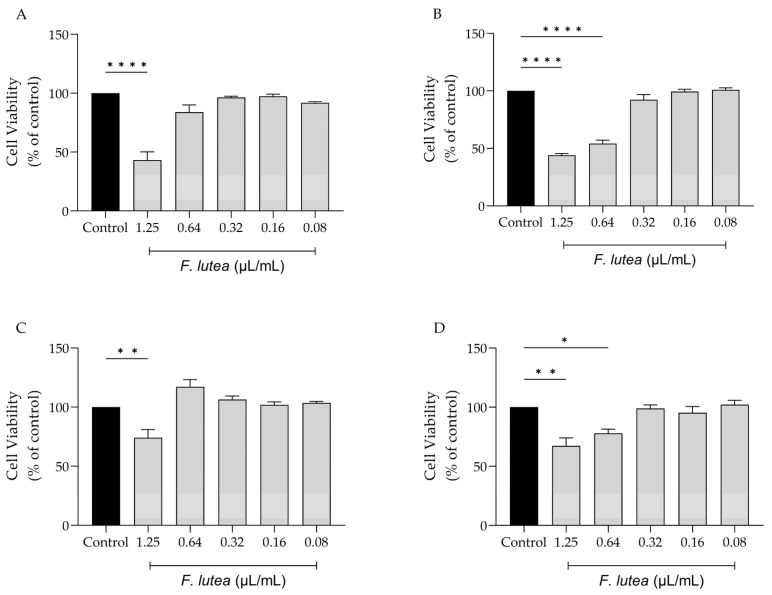
Effect of *F. lutea* essential oil on cell viability. Cell viability was assessed using resazurin metabolization assay. Effect on RAW 264.7 (**A**), NIH/3T3 (**B**), B16V (**C**) and HaCaT (**D**) cell lines. Results show the mean ± SEM of at least three independent experiments performed in duplicate. * *p* < 0.05, ** *p* < 0.01, **** *p* < 0.0001, when compared to control (black bars) after one-way ANOVA followed by Tukey’s multiple comparison test.

**Figure 2 plants-12-03741-f002:**
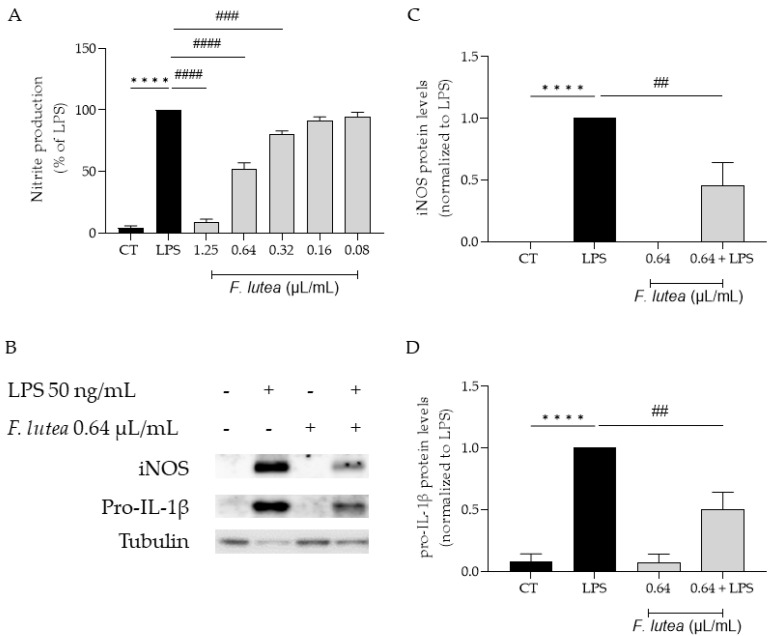
Effect of *F. lutea* essential oil on LPS-stimulated macrophages. (**A**) Nitric oxide production quantified as nitrites in the culture medium using the Griess reaction, after cell treatment with 1.25–0.08 µL/mL of the EO for 1 h followed by stimulation with 50 ng/mL of LPS for 24 h. **** *p* < 0.0001, when compared to control; ^###^ *p* < 0.001; ^####^ *p* < 0.0001, when compared to LPS, after one-way analysis of variance (ANOVA) followed by Dunnett’s multiple comparison test. (**B**) Representative Western blots for iNOS and pro-IL-1β. (**C**) iNOS protein levels. (**D**) Pro-IL-1β protein levels. Tubulin was used as loading control and values were normalized to LPS. Results represent mean ± SEM of at least three independent experiments. **** *p* < 0.0001, when compared to control; ^##^ *p* < 0.01, when compared to LPS, after one-way ANOVA followed by Tukey’s multiple comparison test.

**Figure 3 plants-12-03741-f003:**
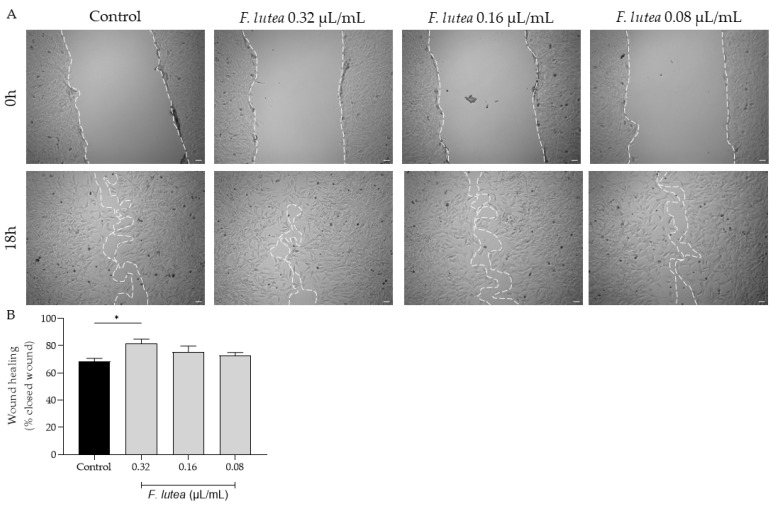
The essential oil from *F. lutea* promotes cell migration. (**A**) Representative bright-field images of NIH/3T3 fibroblasts 0 h and 18 h after scratch in the absence (control) or presence of increasing doses of the essential oil (0.32–0.08 µL/mL). (**B**) Percentage of closed wounds determined using an ImageJ/FIJI plugin. Results represent mean ± SEM of at least three independent assays conducted in duplicate. Scale bar: 50 µm. * *p* < 0.05 when compared to control after one-way ANOVA followed by Tukey’s multiple comparison test.

**Figure 4 plants-12-03741-f004:**
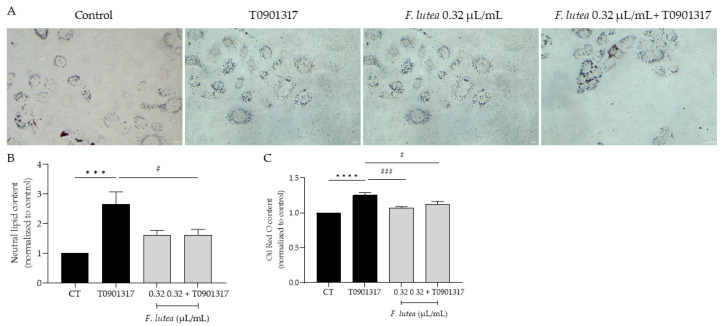
*F. lutea* essential oil modulates lipogenesis in HaCaT keratinocytes. (**A**) Representative bright-field images of HaCaT pre-treated with 0.32 µL/mL of *F. lutea* essential oil followed by 24 h in the presence of T0901317 and then neutral lipids were stained with Oil Red O staining. (**B**) Neutral lipid content in HaCaT keratinocytes quantified using ImageJ/FIJI. (**C**) Neutral lipid content in HaCaT keratinocytes after Oil Red O stain solubilization in isopropanol and absorbance read at 500 nm. A minimum of 5 images were quantified in each replicate. Results shown as mean ± SEM of at least three independent experiments made in duplicate. Scale bar: 50 µm. *** *p* < 0.001, **** *p* < 0.0001 when compared to control, ^#^ *p* < 0.05, ^###^ *p* < 0.001 when compared to T0901317 after one-way ANOVA followed by Tukey’s multiple comparison test.

**Figure 5 plants-12-03741-f005:**
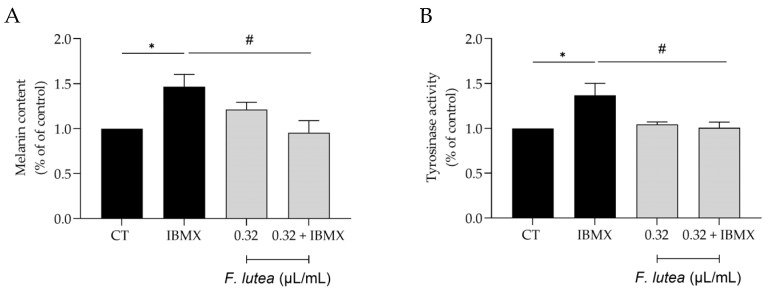
*F. lutea* essential oil exerts a depigmenting effect on B16V melanocytes. (**A**) Melanin content in B16V melanocytes after 48 h of cell treatment with 3-isobutyl-1-methylxantine (IBMX) in the absence or presence of 0.32 µL/mL of *F. lutea* essential oil. (**B**) Tyrosinase activity in B16V melanocytes after 48 h of cell treatment with 3-isobutyl-1-methylxantine (IBMX) in the absence or presence of 0.32 µL/mL of *F. lutea* essential oil. Results show mean ± SEM of at least three independent experiments made in triplicate. * *p* < 0.05 when compared to control, ^#^ *p* < 0.05 when compared to IBMX after one-way ANOVA followed by Tukey’s multiple comparison test.

**Figure 6 plants-12-03741-f006:**
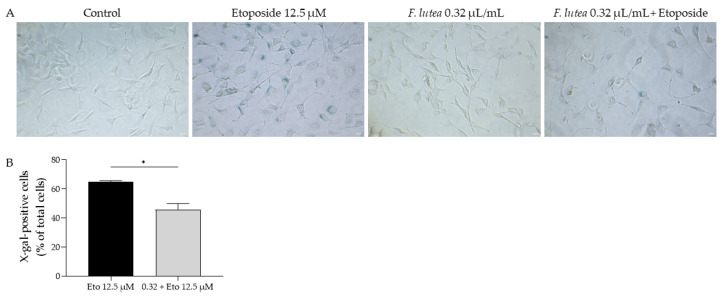
*F. lutea* essential oil decreases senescence-associated β-galactosidase activity. (**A**) Representative bright-field images of NIH/3T3 fibroblasts treated for 24 h with 12.5 µM etoposide, followed by 24 h in etoposide-free medium in the absence or presence of 0.32 µL/mL of *F. lutea* essential oil. (**B**) Percentage of X-gal positive NIH/3T3 fibroblasts treated for 24 h with 12.5 µM etoposide, followed by 24 h in etoposide-free medium in the absence (black bar) or presence (gray bar) of 0.32 µL/mL of *F. lutea* essential oil. Scale bar: 50 µm. * *p* < 0.05 when compared to Eto 12.5 µM, after Mann–Whitney test.

**Figure 7 plants-12-03741-f007:**
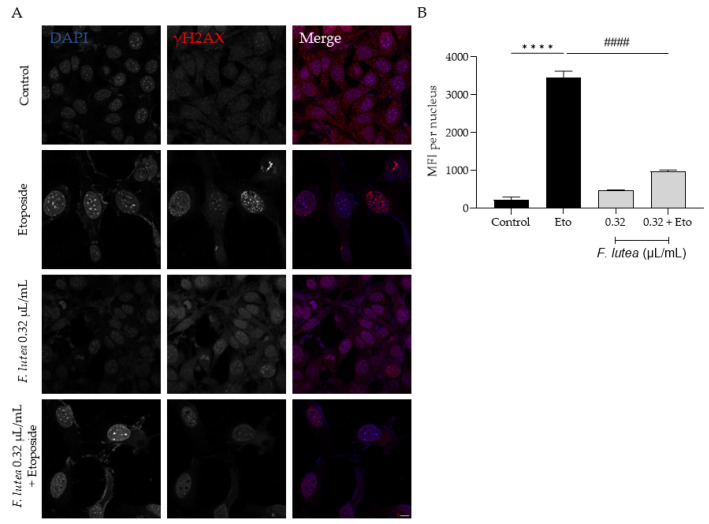
*F. lutea* essential oil decreases the nuclear accumulation of the phosphorylated form of H2AX (γ-H2AX). (**A**) Representative confocal images of NIH/3T3 fibroblasts after 24 h treatment with etoposide (12.5 µM) followed by 24 h in the presence of *F. lutea* essential oil (0.32 µL/mL). γH2AX was stained with Alexa Fluor 564 and nuclei were counterstained with DAPI. (**B**) Mean fluorescence intensity (MFI) per nucleus was quantified using threshold function in ImageJ/Fiji. A minimum of 6 images were used per condition. Results show the mean ± SEM of three independent assays. Scale bar: 10 µm. **** *p* < 0.0001 when compared to control; ^####^ *p* < 0.0001 when compared to Eto.

**Figure 8 plants-12-03741-f008:**
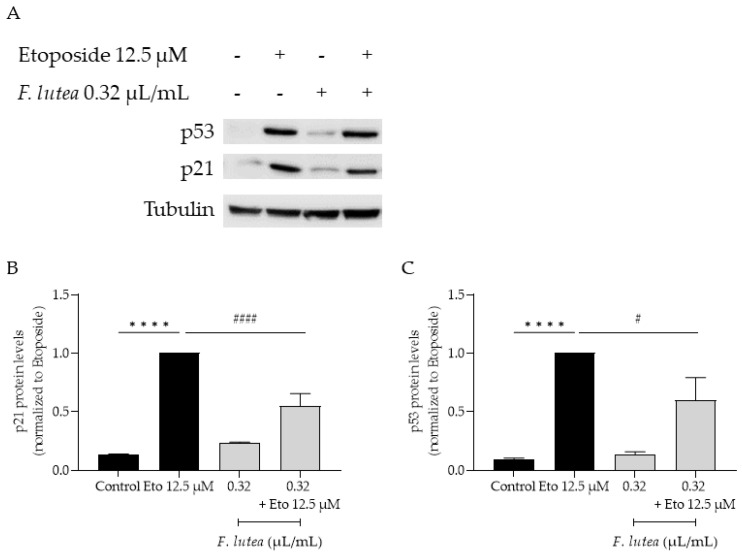
*F. lutea* essential oil modulates p53/p21 signaling pathway. (**A**) Representative Western blot of NIH/3T3 fibroblasts treated with etoposide (12.5 µM) for 24 h followed by an additional 24 h in the absence (black bars) and presence (gray bars) of 0.32 µL/mL of *F. lutea* essential oil. (**B**) Relative expression of p21 protein levels. (**C**) Relative expression of p53 protein levels. Tubulin was used as loading control and values were normalized to etoposide. Results represent mean ± SEM of at least three independent assays. **** *p* < 0.0001 when compared to control and ^#^ *p* < 0.05 and ^####^ *p* < 0.0001 when compared to etoposide after one way ANOVA followed by Tukey’s multiple comparison tests.

**Table 1 plants-12-03741-t001:** Main compounds of the essential oil from *Ferulago lutea*.

Compounds *	RI SPB-1 ^a^	RI SW 10 ^b^	% Peak Area
α-Pinene	930	1030	36.5
Camphene	943	1077	1.5
Sabinene	964	1128	0.5
β-Pinene	970	1118	1.5
Myrcene	980	1161	5.0
*p*-Cymene	1009	1271	1.5
Limonene	1020	1206	31.2
β-Phellandrene	1020	1215	5.5
(*Z*)-β-Ocimene	1025	1235	3.5
*(E)*-β-Ocimene	1035	1253	1.0
γ-Terpinene	1046	1249	0.5
(Z)-Linalool oxide	1055	1439	1.0
Terpinolene	1076	1288	1.0
*p*-Cymenene-8-ol	1160	1621	0.5
α-Copaene	1364	1487	0.5
*(E)*-Caryophyllene	1408	1590	0.5
Germacrene-D	1466	1699	0.5
δ-Cadinene	1508	1751	0.6
Total identified			92.8

* Compounds listed in order of elution in the SPB-1 column. ^a^ RI SPB 1: GC retention indices relative to C_9_-C_23_ n-alkanes on the SPB-1 column. ^b^ RI SW 10: GC retention indices relative to C_9_-C_23_ n-alkanes on the Supelcowax-10 column.

## Data Availability

Data will be made available upon request.

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
