# Peer review of "Effect of Ferulago lutea (Poir.) Grande Essential Oil on Molecular Hallmarks of Skin Aging"

_plants, 2023, doi:10.3390/plants12213741_

Round 1

Reviewer 1 Report

Comments and Suggestions for Authors

The paper deals on the biological activities of the essential oil of aerial parts of Ferulago lutea from Portugal useful to reduce the signs of skin ageing, particularly inflammation, impaired wound healing, pigmentation and senescence, for which no previous scientific studies were previously reported.

The contents of the manuscript are within the scope of the journal and, in general, the experimental design is appropriate and the methodology used for the characterization of the essential oil, and for the different bioassays involving cell viability, anti-inflammatory activity, cell migration, lipogenesis, depigmentation, and anti-senescence properties, is adequate and rigorously performed. Tables and figures correctly support results.

It is noteworthy that the results reported are original and relevant, and should be of profit for the pharmaceutical and cosmetic industries.

The authors have done a complete study on the subject, with exhaustive revision of the recent publications on the topic investigated. The manuscript is well organized and correctly presented.

Nevertheless, some minor considerations should be taken in account before acceptation:

- Line 343: It should be “4.1. Plant material and essential oil distillation”.

- Lines 344 and 345: Concerning the plant material authors write: “Aerial parts (umbels with mature seeds) of F. lutea were collected in the flowering stage in …...”. It is not clear if they collected all the aerial part including stems, leaves and umbels with flowers, or only the latter. On the other hand, at the flowering stage umbels should have mainly flowers and scarcely fruits with seeds. In my opinion, these aspects concerning the plant material should be specified, as well as the date of collection (month and year). On the other hand, they should indicate how was the identification done: did any of the authors or a botanical authority correctly identify the plant material?

So, in my opinion the manuscript describes an original research of high scientific quality with interesting results. It fits with the general aims and scopes of Plants and it can be accepted for publication after some minor revision.

Author Response

The paper deals on the biological activities of the essential oil of aerial parts of Ferulago lutea from Portugal useful to reduce the signs of skin ageing, particularly inflammation, impaired wound healing, pigmentation and senescence, for which no previous scientific studies were previously reported.

The contents of the manuscript are within the scope of the journal and, in general, the experimental design is appropriate and the methodology used for the characterization of the essential oil, and for the different bioassays involving cell viability, anti-inflammatory activity, cell migration, lipogenesis, depigmentation, and anti-senescence properties, is adequate and rigorously performed. Tables and figures correctly support results.

It is noteworthy that the results reported are original and relevant, and should be of profit for the pharmaceutical and cosmetic industries.

The authors have done a complete study on the subject, with exhaustive revision of the recent publications on the topic investigated. The manuscript is well organized and correctly presented.

Nevertheless, some minor considerations should be taken in account before acceptation:

- Line 343: It should be “4.1. Plant material and essential oil distillation”.

We corrected the title of section 4.1. as suggested by the reviewer.

- Lines 344 and 345: Concerning the plant material authors write: “Aerial parts (umbels with mature seeds) of F. lutea were collected in the flowering stage in …...”. It is not clear if they collected all the aerial part including stems, leaves and umbels with flowers, or only the latter. On the other hand, at the flowering stage umbels should have mainly flowers and scarcely fruits with seeds. In my opinion, these aspects concerning the plant material should be specified, as well as the date of collection (month and year). On the other hand, they should indicate how was the identification done: did any of the authors or a botanical authority correctly identify the plant material?

We acknowledge the reviewer’s comment, as recommended, we added the time of collection of the plant material as well as the maturation stage of the umbels. Furthermore, the name of the taxonomist who performed the species identification was added to the manuscript.

So, in my opinion the manuscript describes an original research of high scientific quality with interesting results. It fits with the general aims and scopes of Plants and it can be accepted for publication after some minor revision.

Reviewer 2 Report

Comments and Suggestions for Authors

Dear editor

The manuscript adresses the assessment of the potential antiaging effects of the essential oil extracted from Ferulago lutea (Poir.) Grande collected in Portugal against skin aging hallmarks, such as inflammation, wound healing, pigmentation, and senescence. The hypothesis of antiaging activity designed by the authors is based on previous reports of anti-inflammatory effects through the modulation of the NF-κB pathway. The methodology used is robust and well designed to achieve the proposed objectives. Thus, the manuscript provides relevant information and provides an important advance in knowledge on the topic covered in the research.. Only minor revisions should be carried out before final acceptance of the manuscript for publication:

1)    In Figures 3, 4 and 7, the extension (µm) of the magnification bar must be added to the caption of the photomicrographs.

Author Response

Dear editor

The manuscript adresses the assessment of the potential antiaging effects of the essential oil extracted from Ferulago lutea (Poir.) Grande collected in Portugal against skin aging hallmarks, such as inflammation, wound healing, pigmentation, and senescence. The hypothesis of antiaging activity designed by the authors is based on previous reports of anti-inflammatory effects through the modulation of the NF-κB pathway. The methodology used is robust and well designed to achieve the proposed objectives. Thus, the manuscript provides relevant information and provides an important advance in knowledge on the topic covered in the research.. Only minor revisions should be carried out before final acceptance of the manuscript for publication:

  • In Figures 3, 4 and 7, the extension (µm) of the magnification bar must be added to the caption of the photomicrographs.

We acknowledge the reviewer’s comment, as suggested we have added the extension of the scale bar in all the figures with photomicrographs.